# The Effect of Subclinical Ketosis on the Peripheral Blood Mononuclear Cell Inflammatory Response and Its Crosstalk with Depot-Specific Preadipocyte Function in Dairy Cows

**DOI:** 10.3390/ani14131995

**Published:** 2024-07-06

**Authors:** Hunter R. Ford, Ty M. Mitchell, Tanner Scull, Oscar J. Benitez, Clarissa Strieder-Barboza

**Affiliations:** 1Department of Veterinary Sciences, Davis College of Agricultural Sciences and Natural Resources, Texas Tech University, Lubbock, TX 79409, USA; hunter.ford@ttu.edu (H.R.F.); mitc1167@msu.edu (T.M.M.); tanner.scull@ttu.edu (T.S.); obenitez@ttu.edu (O.J.B.); 2School of Veterinary Medicine, Texas Tech University, Amarillo, TX 79106, USA

**Keywords:** dairy, PBMC, adipose, inflammation, ketosis

## Abstract

**Simple Summary:**

In dairy cows, metabolic stress at the start of lactation can lead to excessive fat mobilization from the body’s fat deposits. This increase in circulating fats is often accompanied by an increase in ketones and is associated with reduced performance and a dysfunctional immune response. However, the relationship between the fat cells responsible for mobilizing fats and the circulating immune cells has yet to be fully understood in these early lactation dairy cattle. To this aim, we isolated and characterized the circulating immune cells from the blood of healthy and hyperketonemic dairy cattle at the beginning of lactation and assessed the effect of the secretions of these immune cells on fat cells from healthy dairy cows. Overall, the immune cells from healthy and metabolically stressed dairy cattle had similar inflammatory profiles at the gene and cytokine level; however, the secretions from these immune cells had distinct effects on the expression of certain genes and on the ability of fat cells to accumulate fats. Together, these results emphasize the importance of additional factors produced by immune cells in metabolically stressed dairy cattle in regulating fat cell function and provide insight into potential mechanisms underlying the development of metabolic stress.

**Abstract:**

During the periparturient period, cows undergo heightened energy demands at lactation onset, paired with reduced dry matter intake, leading to negative energy balance (NEB). Excessive lipolysis-driven adipose tissue remodeling, triggered by NEB, significantly contributes to ketosis in periparturient dairy cows. However, the role of peripheral blood mononuclear cells (PBMCs) in the pathogenesis of ketosis and in modulating adipose tissue function remains poorly understood. Here, we investigated how ketosis affects the transcriptional profile and secretome of PBMCs and its influence on preadipocyte function in visceral adipose tissue (VAT) and subcutaneous adipose tissue (SAT). Twenty-one postpartum Holstein dairy cows were categorized as either subclinical ketosis (SCK; BHB ≥ 1.0 mM) or control (CON; BHB < 0.8 mM) based on blood beta-hydroxybutyrate (BHB) concentration screening. Blood samples were collected intravenously for the isolation of PBMCs and serum metabolic profiling. Ketosis elevated circulating NEFA and BHB levels but reduced total WBC and neutrophil counts. Isolated PBMCs were evaluated for gene expression and used to produce conditioned media (PBMC-CM), during which PBMCs were stimulated with 10 ng/mL LPS. The overall phenotype of PBMCs was largely consistent between SCK and CON cows, with minimal differences detected in immunomodulatory cytokine expression and PBMC-CM composition following stimulation. Preadipocytes isolated from non-ketotic cows were treated with PBMC-CM to assess the effect of PBMC secretomes on adipose cell function. Preadipocytes treated with SCK PBMC-CM showed reduced lipid accumulation compared to those treated with CON PBMC-CM regardless of the depot. SAT preadipocytes had heightened expression of lipid metabolism-related genes, including DGAT1, LIPE, and FASN, compared to VAT when treated with SCK PBMC-CM. Preadipocytes treated with CM from PBMC stimulated by LPS exhibited upregulation in IL1B and IL6 regardless of the depot or source of PBMCs. Together, these results indicate that although PBMC profiles showed minimal differences, preadipocytes treated with PBMC-CM may be influenced by additional factors, leading to altered preadipocyte function and gene expression that may contribute to adipose cellular dysfunction.

## 1. Introduction

Periparturient cows undergo a sudden surge in energy demands triggered by the initiation of lactation and a concurrent decrease in voluntary dry matter intake, leading to a negative energy balance (NEB) [1]. Control of metabolism during the transition from pregnancy to lactation in dairy cows involves numerous homeostatic and homeorhetic regulatory mechanisms. The most significant homeorhetic adaptation to NEB is the increased mobilization of fat from adipose tissue and the release of non-esterified fatty acids (NEFAs) into the bloodstream [2]. Hormonal and endocrine shifts, including increased growth hormone and catecholamines and decreased insulin, heighten the metabolic state during the periparturient period, favoring lipolysis over lipogenesis, regardless of the energy balance status [3]. Lipolysis significantly reduces adipose tissue mass, often exceeding 30% of its volume or weight, initiating a remodeling process characterized by inflammatory responses with immune cell infiltration in transition cows [4]. Continuous and exacerbated lipolysis also increases the partial oxidation of NEFA to ketone bodies by the liver, thus increasing the risk of ketosis [5]. While the maladaptation of adipose tissue to NEB plays a key role in the development of ketosis, how distinct adipose tissue depots influence these responses and its cellular interplay with immune cells in periparturient dairy cows remains little understood.

The adipose tissue surrounding abdominal viscera, known as visceral adipose tissue (VAT), differs from subcutaneous adipose tissue (SAT) in structure, cellularity, and function. Compared to SAT, VAT is composed of smaller adipocytes and has a reduced adipogenic capacity, accompanied by an enhanced pro-inflammatory response to metabolic diseases [6,7]. Underlying these depot-specific differences in dairy cattle are a higher abundance of pro-inflammatory immune cells in VAT and mature adipocytes with pro-inflammatory and suppressed lipogenic transcriptional profiles [8]. While cellular composition and the transcriptomic profile of SAT of ketotic and non-ketotic cows seem to differ minimally, VAT from cows with subclinical ketosis contains a more pro-inflammatory profile, indicating a potential link between VAT dysfunction and the pathogenesis of metabolic disease in periparturient dairy cattle [9]. Although significant research has targeted characterizing the functional differences between VAT and SAT, we lack a comprehensive understanding of how these differences extend to their interactions with immune cells.

The relationship between nutrient availability and immune cell function has spurred research into bovine immune cell functional and transcriptional profiles during the transition period. Circulating peripheral blood mononuclear cells (PBMCs) are exposed to variations in fluid composition, including fluctuations in circulating nutrients, substrates, and hormones [10] and are reflective of the host immune system. Thus, characterizing PBMC profiles offers valuable insight into immune functionality and potential disease biomarkers [11]. Previous investigations have revealed suppressed immune responses in the PBMCs of dairy cattle with subclinical ketosis [12,13], and high circulating NEFA [13,14]; however, none of these prior studies have evaluated the relationships between PBMC profiles and adipose tissue function in the context of ketosis. Although PBMCs function as a valuable reservoir of immune cells that can infiltrate and modulate adipose tissue [15], the crosstalk between these cells and adipose tissue in dairy cows remains a gap in our knowledge.

Our objective was to define changes in PBMC immunomodulatory cytokine gene expression profiles and secretomes in cows with and without ketosis and how these changes affect VAT and SAT preadipocyte function. Using a combination of experimental approaches, we hypothesized that PBMCs from dairy cattle with subclinical ketosis exhibit dysregulated gene expression profiles with increased pro-inflammatory cytokine production. In addition, we expected PBMC-conditioned media to inhibit proliferation and lipid accumulation in preadipocytes, driven by the suppression of key lipogenic genes.

## 2. Materials and Methods

### 2.1. Animals and Experimental Design

The Institutional Animal Care and Use Committee (IACUC) of Texas Tech University approved all the procedures for this study (protocol no. 21024-04). An overview of the experimental design is provided in Figure 1. A total of 21 multiparous postpartum Holstein dairy cows (3–16 days in milk-DIM) were sourced from two commercial dairy farms and screened for ketosis based on blood beta-hydroxybutyrate (BHB) concentrations using a Precision Xtra (Abbott Laboratories, Green Oaks, IL, USA). Cows were allocated into two groups based on blood BHB: subclinical ketosis (SCK; BHB ≥ 1.0 mM; *n* = 11) or non-ketotic/control cows (BHB ≤ 0.8 mM; *n* = 10). Cows were blocked using DIM, number of lactations, and body condition score (BCS), as shown in Table 1. The body condition score was assessed by two experienced people using a 1–5 scale with 0.25 variations [16]. All of the samples were collected and processed on the same day on each farm. The commercial dairy source of cows did not have a significant effect on any of the serum metabolic parameters, white blood cell counts, or genes assessed via RTqPCR in the isolated PBMCs. Cows were housed in free stalls and received a standard transition diet.

### 2.2. Blood Collection and Serum Metabolic Analysis

Intravenous blood was obtained via coccygeal venipuncture from the selected dairy cows. Blood samples were collected into two 10 mL tubes containing EDTA (Becton Dickinson and Company, Franklin Lakes, NJ, USA, Cat# 366643) and kept on ice and one 10 mL tube (Becton Dickinson and Company, Cat# 368045) for obtaining serum samples. One EDTA tube was utilized for the white blood cell count and differential, and the other tube was used for PBMC isolation. Serum fractions were collected after centrifugation for 10 min at 900× *g*, aliquoted, and stored in a −80 °C freezer for metabolic profiling at the Texas A&M Veterinary Medical Diagnostic Laboratory (College Station, TX, USA).

### 2.3. White Blood Cell Count

White blood cell counts and differentials were performed using a IDEXX Procyte DX hematology analyzer (IDEXX Laboratories, Inc., Westbrook, ME, USA) for absolute counts of total white blood cells, monocytes, neutrophils, lymphocytes, as well as the relative percentages of neutrophils, lymphocytes, and monocytes.

### 2.4. PBMC Isolation

Peripheral blood mononuclear cells were isolated using a modified protocol based on previous methods [17,18]. Briefly, whole blood samples collected in EDTA tubes were centrifuged for 10 min at 900× *g*. Following centrifugation, the buffy coat was pipetted off into a new 15 mL conical tube and diluted 1:1 with 1X phosphate-buffered saline (PBS) (ThermoScientific, Waltham, MA, USA, Cat# BP3994) containing 2% fetal bovine serum (FBS) (Corning, Corning, NY, USA, Cat# 35-016-CV). The buffy coat/PBS mixture was then slowly pipetted into a new 15 mL conical tube containing 5 mL of Ficoll reagent (Cytiva, Marlborough, MA, USA Cat# 17144002). The tube was then centrifuged for 22 min at 900× *g*, after which the newly separated PBMC layer was pipetted into a new 15 mL conical tube with 10 mL of PBS + 2% FBS and centrifuged for 5 min at 250× *g*. This step was repeated an additional 1–2 times until the supernatant was no longer cloudy. Finally, the supernatant was discarded and the PBMC pellet was resuspended in preadipocyte media. A small subsample (20 μL) from each PBMC isolate was stained with 0.4% trypan blue (Gibco, Waltham, MA, USA, Cat# 15250-061) and counted using an automated cell counter (Countess 3, Life Technologies Inc., Carlsbad, CA, USA) to determine cell concentration. All PBMC samples had viability >85%.

### 2.5. Preparation of Conditioned Media

Peripheral blood mononuclear cells isolated from Control and SCK cows were transferred to 24-well plates at a density of 1 × 10^6^ cells/well in preadipocyte media containing Dulbecco’s modified Eagle’s medium F12 50:50 (Corning, Cat# 10-090-CV) supplemented with 10% FBS (Corning, Cat# 35-016-CV), 1% (*v*/*v*) antibiotic–antimycotic (Gibco, Cat# 15240-062), 100 μM ascorbic acid (Sigma-Aldrich, Cat# A4544-25G), 33 μM biotin (Sigma-Aldrich, Burlington, MA, USA, Cat# B4639-500mg), 17 μM D-Pantothenate (ThermoScientific, Cat# 243305000), and 20 mM HEPES (ThermoScientific, Cat# J16924.K2). Half of the wells containing PBMCs isolated from each animal were stimulated with 10 ng/mL LPS (Invitrogen, Waltham, MA, USA, Cat# 00-4976-93) and cultured for 24 h at 37C/5% CO_2_. After 24 h, the media was removed from each well and saved for use as “conditioned media” in other experiments. Conditioned media collected from PBMCs isolated from the Control cows and treated with LPS was denoted as Control Stimulated (CS), while that not treated with LPS was denoted as Control Non-Stimulated (CN). Conditioned media collected from PBMCs that were isolated from ketotic cows and treated with LPS was denoted as Ketosis Stimulated (KS), while that collected from wells not treated with LPS was denoted as Ketosis Non-Stimulated (KN). To assess the effects of ketosis on the crosstalk between PBMC and preadipocyte function, we utilized pooled PBMC-conditioned media from SCK vs. control animals.

### 2.6. PBMC Cytokine Assessment

Subsamples from conditioned media from each cow, as well as pooled conditioned media samples, were assayed for IL-6, TNFα, and IFNγ concentrations via bovine DuoSet^®^ ELISA kits (R&D Systems Inc., Minneapolis, MN, USA; IL-6: Cat# DY8190; TNF-α: Cat# DY2279, and IFN-γ: Cat# DY2300) following the manufacturer’s protocol.

### 2.7. Adipose Tissue Collection and Digestion

Preadipocytes used for the proliferation, lipid accumulation, and RTqPCR experiments were isolated from the adipose tissue of four clinically healthy (non-ketotic) postpartum Holstein dairy cows (7.0 ± 1.4 DIM; 3.0 ± 0.8 lactations; 3.7 ± 0.2 BCS) with a similar metabolic profile to the Control cows used for PBMC isolation (Appendix A). Abdominal subcutaneous adipose tissue (SAT) and retroperitoneal visceral adipose tissue (VAT) were collected from each animal via laparotomy. Briefly, the right flank fossa was clipped and scrubbed using chlorhexidine and 70% alcohol thrice. Then, 20 mL of 2% lidocaine hydrochloride (VetOne, Boise, ID, USA, Cat# 510213) was applied in an inverted L block, and a 10 cm skin incision was made 10 cm caudally and parallel to the last rib, and 7–10 cm below the costal junction. Then, 5 to 10 g of SAT was obtained from the flank incision. Next, muscles of the abdominal wall were incised and dissected until reaching the peritoneum. A 20 g sample was obtained from the retroperitoneum fat. Muscle layers were closed with a simple continuous suture pattern using synthetic absorbable violet-coated braided polyglactin 910—USP 3 + 4 (Riverpoint Medical), while the skin incision was closed with a Ford interlocking suture pattern using non-absorbable polyamide thread, pseudo-monofilament suture USP 3 (Braunamid, Loveland, CO, USA, Cat# J009103). Adipose tissue samples used for digestion were kept in modified Krebs–Ringer buffer [8] at 37C during transport and prior to digestion. The digestion of adipose tissue samples for isolation of the stromal vascular fraction (SVF) and expansion and adipogenic induction of preadipocytes in vitro were performed as previously described by our laboratory [8].

### 2.8. Cell Culture

Preadipocytes were grown and cultured in preadipocyte media with media replacement every 48 h. To differentiate confluent preadipocytes, induction media, composed of preadipocyte media supplemented with 0.5 μg/mL of insulin (Sigma-Aldrich, Cat# 10516-5ML), 10 mM of acetate (Sigma-Aldrich, Cat# 3863-50ML), 1 mM octanoate (ACROS Organics, Waltham, MA, USA, Cat# A0413228), 5 μM troglitazone (AdipoGen Life Sciences, San Diego, CA, USA Cat#502053932), 10 μM ciglitazone (AdipoGen, Cat#502053787), 5 μg/mL transferrin (Sigma-Aldrich, Cat# T1283), 0.5 mM 3-isobutyl 1-methyxanthine (IBMX; AdipoGen Life Sciences, Cat# AG-CR1-3512-G001), and 1 μM dexamethasone (Sigma-Aldrich, Cat# D2915-100MG), was used for 48 h. After 48 h, maintenance media, composed of induction media without IBMX and dexamethasone, was used for 12 days with media replacement every 48 h. All of the cells were cultured in an incubator at 37C/5% CO_2_.

### 2.9. Preadipocyte Proliferation and Lipid Accumulation

Preadipocytes were plated in preadipocyte media in 96-well plates at a density of 2000 cells/well. After 48 h, the media was removed and replaced with either CS, CN, KN or KS-conditioned media. The treatment media was replaced every 48 h for the duration of the proliferation assay. Preadipocyte proliferation was assessed via confluency measurements using Gen5 Image Prime software (version 3.14) in a Cytation 5 multi-mode plate reader (Biotek, Santa Clara, CA, USA). The imaging of all wells was performed at the same well location once a day every 48 h for 7 days. Utilizing a 4X objective, one brightfield image per quadrant of the well was captured. To enhance data accuracy, the images underwent processing to diminish background noise and enhance contrast. Following the last measurement of proliferation on day 7, preadipocytes were induced to differentiate into adipocytes, as described above. After 14 days of differentiation, adipocytes were assayed for lipid accumulation using the AdipoRed™ reagent (Lonza Biosciences, Walkersville, MD, USA, Cat# T-7009) in accordance with the manufacturer’s instructions using the Cytation 5 multi-mode plate reader. Fluorescence values were corrected by the number of cells in each well that were counted using Cytation 5 prior to the AdipoRed™ assay.

### 2.10. Preadipocyte-PBMC Crosstalk

We studied the effect of PBMC secretome on preadipocyte adipogenic, lipogenic, and inflammatory gene expression by exposing cells to PBMC-conditioned media. Briefly, paired VAT and SAT preadipocytes from the same postpartum non-ketotic cows were plated in preadipocyte media in 6-well plates at a density of 100,000 cells/well. Preadipocyte media was replaced every 48 h until cells reached 80% confluency. Next, preadipocyte media was replaced with either CS, CN, KN, or KS PBMC-conditioned media. After 48 h of treatment, the media was removed, preadipocytes were rinsed with cold 1X PBS, and nucleic acids were collected using RLT buffer (Qiagen, Germantown, MD, USA, Cat# 1015762) for downstream RTqPCR analysis.

### 2.11. RTqPCR

We performed RTqPCR in isolated PBMC samples from each cow, as well as in VAT and SAT preadipocytes treated with pooled CS, CN, KN, or KS PBMC-conditioned media. For the isolated PBMCs, 1 × 10^6^ cells were suspended in RLT buffer directly after PBMC isolation. For both preadipocytes and isolated PBMCs, RNA was extracted using the RNeasy Mini Kit (Qiagen, Cat# 74104), following the manufacturer’s instructions. RNA concentration and purity were assessed using a NanoDrop One© Microvolume UV-Vis Spectrophotometer (ThermoScientific, Waltham, MA, USA). Following RNA isolation, cDNA was prepared using the High-Capacity cDNA Reverse Transfection Kit (Applied Biosystems, Cat# 4368814) in a MiniAmpPlus thermal cycler (Applied Biosystems, Waltham, MA, USA). RTqPCR was performed using TaqMan^®^ gene expression assays and reagents (Life Technologies Inc., Carlsbad, CA, USA) in a QuantStudio 6 Pro (Applied Biosystems, Waltham, MA, USA). Target genes for the preadipocyte samples were diacylglycerol o-acyltransferase 1 (*DGAT1*; Bt03251717_g1), fatty acid synthase (*FASN*; Bt0310481_m1), hormone-sensitive lipase (*LIPE*; Bt0323697_m1), perilipin 1 (*PLIN1*; Bt03257414_m1), peroxisome proliferator-activated receptor gamma (*PPARG*; Bt03217547_m1), transforming growth factor beta 1 (*TGFB1*; Bt04259484_m1), interleukin-1 beta (*IL1B*; Bt03212741_m1), and interleukin 6 (*IL6*; Bt03211905_m1). Target genes for the PBMC samples were colony-stimulating factor 2 (*CSF2*; Bt03212483_m1), interleukin 8 (*CXCL8*; Bt03211906_m1), interferon gamma (*IFNG*; Bt03212723_m1), interleukin 10 (*IL10*; Bt03212727_m1), interleukin 17A (*IL17A*; Bt03210251_m1), interleukin 18 (*IL18*; Bt03212732_m1), *IL1B* (Bt03212741_m1), interleukin 4 (*IL4*; Bt03211897_m1), *IL6* (Bt03211905_m1), *TGFB1* (Bt04259484_m1), and tumor necrosis factor alpha (*TNF*; Bt03259156). Beta-2 microglobulin (*B2M*; Bt03251628_m1) and eukaryotic initiation factor 3 subunit K (*EIF3K*; Bt03226565) were used as housekeeping genes [19,20] for the preadipocyte samples, while beta-actin (*ACTB*; Bt03279174_g1) and 40S ribosomal protein S9 (*RPS9*; Bt03272016) were used as housekeeping genes for the PBMC samples [21]. Prior to analysis, expression data were normalized to the geometric mean of the expression of the housekeeping genes. For the PBMCs, data are expressed as fold change relative to expression in PBMCs from Control cows for each gene.

### 2.12. Statistical Analysis

All of the statistical analyses were performed using the PROC GLM method in SAS (version 9.4; SAS Institute Inc., Cary, NC, USA). The normality of the residuals was assessed using the PROC UNIVARIATE method in SAS (v9.4), and none of the data required transformation prior to analysis. For metabolic profile markers, white blood cell counts, and PBMC RTqPCR data, group (Control, SCK) was used as the fixed effect, with cow as the random effect. For ELISAs, LPS treatment (+LPS, −LPS), and the group that the PBMCs used to make the conditioned media came from (Control, SCK), were used as the fixed effects, with cow as the random effect. For preadipocyte proliferation, lipid accumulation, and preadipocyte RTqPCR data, LPS treatment (+LPS, −LPS), the group that the PBMCs used to make the conditioned media came from (Control, SCK), and adipose tissue depot (SAT, VAT) were used as the fixed effects, with the cow that the preadipocytes were derived from as the random effect. Outliers with Studentized *t* values ≥ 3.0 were removed from the analysis. Significance was declared at *p* ≤ 0.05, with tendencies declared for 0.05 < *p* ≤ 0.1.

## 3. Results

### 3.1. Subclinical Ketosis Increased Circulating NEFA and BHB, but Decreased Total WBC and Neutrophils

Serum metabolites were quantified to assess the metabolic profile associated with postpartum and ketosis in the experimental cows. As anticipated, SCK cows exhibited elevated concentrations of circulating BHB (*p* < 0.01; Figure 2A) and nonesterified fatty acids (NEFA; *p* = 0.02; Figure 2B) compared to the Control. There were no differences between SCK and Control groups in terms of glucose, insulin, cholesterol, blood urea nitrogen (BUN), albumin, Ca, P, Mg, Na, K, or Cl serum concentrations (Appendix A). Analysis of white blood cell (WBC) populations from whole blood samples revealed that SCK cows had lower counts of both total WBC (*p* = 0.01; Figure 2C) and total neutrophils (*p* = 0.02; Figure 2C) compared to the Control cows. However, the relative abundance of different types of WBCs was not different between the two groups, with only the percentage of monocytes tending to be higher in SCK cows (Figure 2D).

### 3.2. Subclinical Ketosis Had a Minimal Effect on PBMC Expression of Target Genes

No significant differences were detected in the expression of the selected target genes in the PBMCs from SCK and Control cows (Figure 3A). There were tendencies for higher IL18 and IL6 expression in PBMCs from SCK cows.

### 3.3. LPS Stimulation, but Not SCK, Affects IL-6 and TNFα Concentrations in PBMC-Conditioned Media

To investigate the inflammatory secretome of PBMCs under both basal and inflammatory conditions, enzyme-linked immunosorbent assays were conducted to quantify the pro-inflammatory cytokines TNFα, IL-6, and IFNγ (Figure 3B). The source of PBMCs (Control vs. SCK) had no effect on the amount of IL-6 or TNFα secreted into the media; however, both cytokines were increased when PBMCs were stimulated with LPS. There were no effects on IFNγ concentrations in the PBMC-conditioned media.

### 3.4. Preadipocytes Exposed to SCK PBMC-Conditioned Media Accumulate Less Lipids

We assessed the effect of a 7-day exposure of VAT and SAT preadipocytes to PBMC-conditioned media on adipogenesis by measuring preadipocyte proliferation and adipocyte lipid accumulation. Preadipocyte proliferation increased throughout 7 days, with VAT cells proliferating at a lower rate compared to SAT (Figure 4A). SCK tended to increase preadipocyte proliferation in both depots, while LPS stimulation tended to decrease proliferation (Figure 4A). As shown in Figure 4B, adipocyte lipid accumulation was affected by the interaction between depot, LPS stimulation, and whether the PBMC-conditioned media derived from SCK or Control cows. VAT cells treated with CN-conditioned media had the highest lipid accumulation, and SAT cells treated with KS-conditioned media had the lowest. Additionally, there was a higher lipid accumulation in cells treated with conditioned media from Control PBMCs compared to conditioned media from SCK PBMCs (*p* = 0.03; Figure 4B).

### 3.5. Conditioned Media from SCK PBMCs Increases Expression of Genes Associated with Lipid Metabolism in Preadipocytes

The expressions of genes associated with lipid metabolism in preadipocytes treated with PBMC-conditioned media for 48 h are shown in Figure 5. The expression of *DGAT1* was higher in SAT preadipocytes compared to VAT preadipocytes (Figure 5A) and also higher in preadipocytes treated with SCK PBMC-conditioned media (Figure 5A). There was an interaction between the depot and source of conditioned media, with SCK PBMC-conditioned media having a stronger stimulatory effect on *DGAT1* expression in SAT preadipocytes compared to VAT preadipocytes (Figure 5A). There was a tendency for an interaction effect of depot × LPS, with higher *DGAT1* expression in SAT preadipocytes compared to VAT preadipocytes when the cells were treated with PBMC-conditioned media stimulated with LPS. A tendency for an interaction effect of LPS*SCK was also observed, with KS-conditioned media increasing *DGAT1* expression more than CS-conditioned media. Following the same trends as *DGAT1*, *LIPE* expression was also higher in SAT compared to VAT, in preadipocytes treated with SCK PBMC-conditioned media, with a stronger effect of SCK PBMC-conditioned media in SAT compared to VAT (Figure 5B). Both *FASN* (Figure 5C) and *PLIN1* (Figure 5D) expressions were higher in preadipocytes treated with SCK PBMC-conditioned media, with a tendency for higher *FASN* expression in SAT preadipocytes compared to VAT preadipocytes. The expression of *PPARG* was not affected by the treatments (Figure 5E); however, there was a tendency for higher *PPARG* expression in preadipocytes treated with SCK PBMC-conditioned media compared to Control PBMC-conditioned media.

### 3.6. Preadipocyte Cytokine Gene Expression Is Affected by LPS and SCK PBMC-Conditioned Media

The RTqPCR results for the expression of *IL1B*, *IL6*, and *TGFB1* in preadipocytes treated with PBMC-conditioned media are shown in Figure 6. The expression of *TGFB1* followed a similar trend to the target genes associated with lipid metabolism, with an increase in *TGFB1* expression upon preadipocyte treatment with SCK PBMC-conditioned media (Figure 6A). The expression of *IL1B* and *IL6* was minimal in preadipocytes treated with non-stimulated conditioned media independent of ketosis status (Figure 6B,C). As expected, LPS exposure increased *IL1B* and *IL6* expression. For *IL6*, expression tended to be higher in SAT preadipocytes compared to VAT preadipocytes. There was a tendency for the interaction effect between depot and LPS exposure on *IL6* expression, with LPS tending to stimulate a stronger stimulatory effect in SAT vs. VAT preadipocytes.

## 4. Discussion

Overall, our findings indicate that subclinical ketosis has a moderate effect on WBC profiles, albeit with minor effects on cytokine gene expression and secretion in PBMCs. Elevated levels of ketones, such as those detected in the subclinical ketotic cows in this study, is relatively common in high-producing dairy cattle and may be more indicative of higher energy demands rather than a pathological condition or metabolic dysfunction [22,23]. Our findings highlighted the effects of PBMC-conditioned media on preadipocyte gene expression and function, suggesting that other factors coming from PBMCs can regulate adipose cell function.

Prior studies on early postpartum dairy cattle describe varying differences in WBC differentials between cows with subclinical ketosis and non-ketotic cows, with some studies observing increases in total WBC [24], lymphocytes [24,25], and monocytes [26] in the blood of dairy cows with subclinical ketosis and no effect on neutrophil populations [25]. Our results, observing lower total WBC in the blood of SCK cows, somewhat contrast with those previous findings, even though the levels of BHB and NEFA in the SCK cows used in this study were similar to those studies. Given that subclinical ketosis has been associated with an increased incidence of other postpartum diseases in dairy cattle [27,28], underlying conditions such as endometritis [29] and subclinical mastitis [30] that have yet to be detected can also influence WBC, contributing to variations between studies. In our study, at the moment of animal selection and sampling, no other clinical diseases were reported or detected, and WBC parameters fell within the reference ranges for early lactation dairy cattle [31]. Also limiting the comparison between our study and previous studies is the absence of inflammatory parameters (ex. haptoglobin, paraoxonase, myeloperoxidase, etc.) in our dataset. Inflammatory conditions in postpartum dairy cows are known to influence immune cell function [32,33,34]; however, the inflammatory status of cows used in our study was not registered or noted. During in vitro studies, elevated concentrations of NEFA [35] (≥0.5 mM) and BHB [36] (≥1.0 mM) suppress the proliferation of PBMCs from dairy cattle; however, these in vitro settings often fail to replicate endogenous factors, such as albumin, which can modulate the effects of serum components on white blood cell function in vivo.

Despite the lower total WBC and neutrophils in the blood of SCK cows, the relative abundance of each WBC type was not different between groups, with only a tendency for a greater % of monocytes in SCK cows. In line with the lack of differences, no significant differences were observed in the expression of target genes in PBMCs, albeit with tendencies for increased *IL6* and *IL18* expression in PBMCs from SCK cows. IL-6 is a multifaceted cytokine involved in coordinating immune and inflammatory responses [37], and increased PBMC expression of genes involved in IL-6 signaling has been identified in metabolically stressed dairy cattle during the early postpartum period [13]. However, the increase in IL-6 signaling may be driven by factors other than NEFA or BHB, as neither of these compounds increased *IL6* expression in bovine PBMCs in vitro [38]. IL-18 is another important pro-inflammatory cytokine involved in stimulating IFNγ production by immune cells [39]; however, we observed no differences in *IFNG* gene expression or IFNγ production between PBMCs from SCK and Control cows. Additional investigations evaluating the relationship between postpartum metabolic stress and *IL18* in dairy cattle are limited. The absence of differences in PBMC *IL6*, *TNFA*, and *IFNG* expression between SCK and Control was also supported by similar concentrations of IL-6, TNFα, and IFNγ measured in the PBMC-conditioned media. In addition, while both PBMC secreted IL-6 and TNFα concentrations increased after LPS treatment, SCK and Control responded similarly to LPS. Together, these data indicate that, among the genes and cytokines measured, the overall phenotype of PBMCs is largely similar between SCK and Control cows. However, findings regarding the effects of PBMC-conditioned media on preadipocyte function and gene expression suggest that other PBMC factors secreted in ketotic conditions can modulate adipose cell function, particularly adipogenesis.

We assessed how PBMC secretory factors associated with ketosis could affect adipogenesis in vitro by evaluating preadipocyte proliferation, the gene expression of adipogenic and lipogenic markers, and adipocyte lipid accumulation. Independent of ketotic status and LPS stimulation, PBMC-conditioned media had no effect during the proliferation assay, with SAT cells having a greater proliferative capacity than VAT cells. Notable, however, was the observation that even after conditioned media had been removed for 14 days, cells that had been treated with the conditioned media from SCK PBMCs had overall reduced lipid accumulation, independent of depot or LPS stimulation, suggesting a detrimental effect of PBMC secretome from SCK on adipocyte function. One potential factor driving this effect may be the higher expression of *TGFB1* in preadipocytes treated with SCK PBMC-conditioned media. TGFB1 is a member of the transforming growth factor beta cytokine family and has been recognized as an important inhibitor of adipogenesis [40]. Prior studies have shown that TGFB1 signaling inhibits the differentiation of both 3T3-L1 cells [41] and porcine preadipocytes [42,43] into mature adipocytes. Furthermore, TGFB1 has been implicated in adipose dysfunction under conditions of obesity, evidenced by the reduced expression of adipogenic marker genes (*ADIPOQ*) in mouse adipocytes treated with TGFB1 [44] and increases in *Tgfb* expression in adipose tissue of obese mice [45]. Thus, the increased expression of *TGFB1* when preadipocytes were treated with conditioned media from PBMCs from SCK cows suggests that factors produced by PBMCs in cows with subclinical ketosis function to inhibit adipogenesis and the differentiation of preadipocytes into mature adipocytes. Although contrary to this is the increase in expression of key lipogenic genes, such as *DGAT1*, *FASN*, *LIPE*, and *PLIN1*, under the same conditions, perhaps an indicator of a more dysregulated adipogenic signaling as opposed to a coordinated decrease in adipogenesis. Previous investigations have identified minimal differences, and even reductions in the expression of important lipogenic genes, such as *FASN*, *PPARG*, and *LIPE*, in early postpartum dairy cows with ketosis [46,47,48], indicating that the increases in the expression of these genes observed in our study may be attributed to PBMC-derived factors. Both *PLIN1* and *LIPE* participate in the regulation of lipogenesis and lipolysis via phosphorylation events, with phosphorylation of both proteins promoting lipolysis. Insulin functions as an inhibitor of these phosphorylation events, preventing the phosphorylation of PLIN1 and LIPE and promoting lipid accumulation and inhibiting lipolysis [49,50]. In contrast, catecholamines, including dopamine, norepinephrine, and epinephrine, stimulate the phosphorylation of PLIN1 and LIPE, thereby promoting lipolysis [51]. Interestingly, catecholamines can be produced not only by the central nervous system but also by PBMCs [52,53,54], and increased phosphorylated LIPE has been observed in the adipose tissue of dairy cattle with subclinical ketosis [55]. Within the context of this investigation, we speculate that increased catecholamine production by PBMCs from SCK cows, combined with the increase in *TGFB1* expression, may be driving the reduction in lipogenic activity we observed in our cultured cells. Notably, the stimulatory effect of SCK PBMC-conditioned media on the expression of *DGAT1*, *LIPE*, and *FASN* was more pronounced in SAT preadipocytes compared to VAT, an effect that may also be attributable to differences in adrenergic sensitivity between the two depots [56]. Prior work by Queathem et al. demonstrated that adrenergic stimulation increased *FASN* expression in SAT and decreased inhibition of AMPK in VAT in mice [57]. AMPK, which is activated upon adrenergic signaling, functions as an important regulator of energy metabolism in adipose tissue, with increased AMPK activity promoting lipolysis and suppressing lipogenesis [58]. Therefore, continued investigations into the production of catecholamines by PBMCs in transition dairy cattle are essential to understanding how these peripheral cells may contribute to postpartum adipose tissue dysfunction.

Finally, we observed an increase in *IL1B* and *IL6* expression in preadipocytes treated with conditioned media stimulated by LPS. These findings are supported by prior research where LPS stimulated an increase in *IL1B* expression in human visceral adipocytes [59], with non-stimulated cells expressing very little *IL1B*. Similar results have been observed for *IL6*, where LPS stimulation of porcine adipocytes triggered an increase in *IL6* expression [60]. The lack of a difference in *IL1B* or *IL6* expression between SAT and VAT preadipocytes aligns with previous studies in dairy cattle as well. Michelotti et al. [8] found no difference in *IL1B* or *IL6* expression between SAT and VAT from dairy cattle. Together, these results indicate that factors derived from PBMCs from either non-ketotic or subclinically ketotic cows have a minimal effect on *IL1B* or *IL6* expression in bovine preadipocytes.

## 5. Conclusions

In conclusion, these results demonstrate that although differences in PBMC cytokine gene expression and secretion are minimal between dairy cattle with and without subclinical ketosis, additional factors secreted by PBMCs may contribute to adipose cellular dysfunction. Notably, the effects of PBMC-conditioned media on adipose cell gene expression occur in a depot-specific manner, further emphasizing the importance of considering depot-specific characteristics in metabolic dysfunction. Further investigations aimed at characterizing the PBMC secretome in early postpartum dairy cattle will provide critical insight into the pathogenesis of ketosis in dairy cattle and may lead to the identification of predictive biomarkers and therapeutic targets.

## Figures and Tables

**Figure 1 animals-14-01995-f001:**
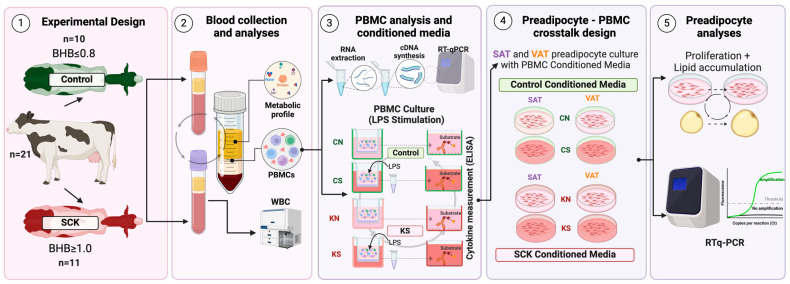
Experimental design describing PBMC collection and analysis, conditioned media and preadipocyte crosstalk experiments. Preadipocytes utilized in the crosstalk experiments were isolated from non-ketotic cows during early lactation (described in Section 2.7). SAT = subcutaneous adipose tissue preadipocytes; VAT = visceral adipose tissue preadipocytes; CN = conditioned media from Control PBMCs without LPS stimulation; CS = conditioned media from Control PBMCs with LPS stimulation; KN = conditioned media from SCK PBMCs without LPS stimulation; KS = conditioned media from SCK PBMCs with LPS stimulation. Created with BioRender.com.

**Figure 2 animals-14-01995-f002:**
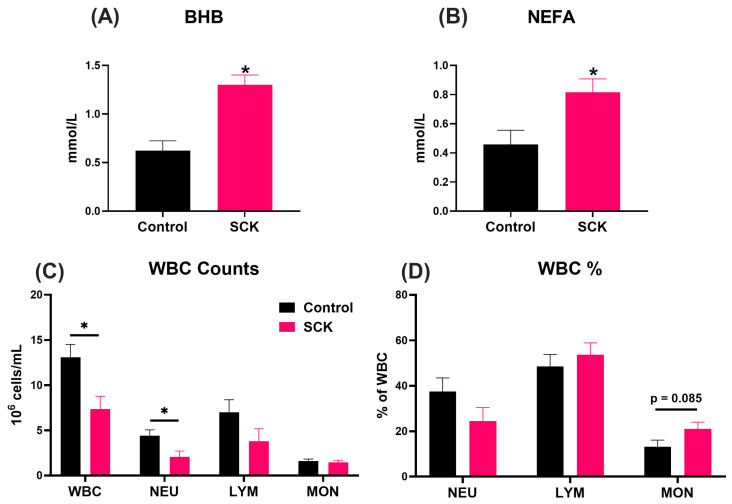
Serum BHB (**A**), NEFA (**B**), and whole blood WBC counts (**C**) and differential (**D**) in Control (*n* = 10) and SCK (*n* = 11) cows. BHB = β-hydroxybutyrate; NEFA = nonesterified fatty acids; WBC = white blood cell; NEU = neutrophils; LYM = lymphocytes; MON = monocytes. Significant differences between groups are denoted by *.

**Figure 3 animals-14-01995-f003:**
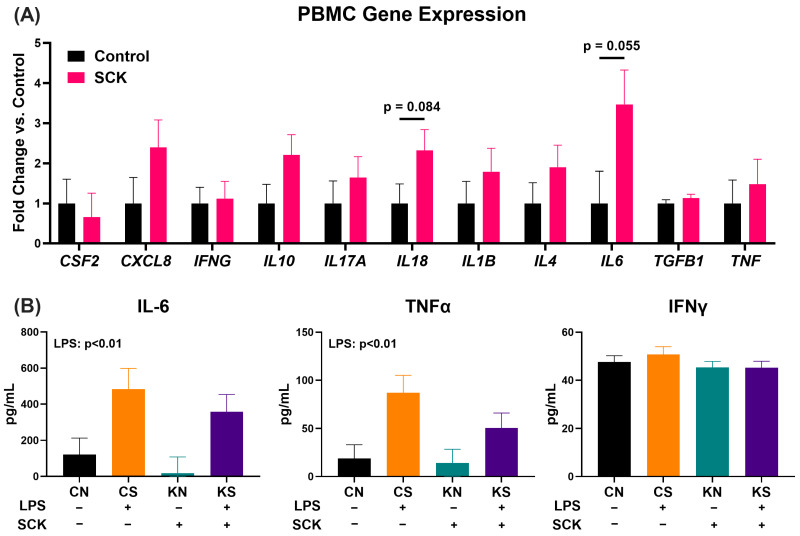
Expression of target genes in PBMCs isolated from Control and SCK cows assessed via RTqPCR (**A**) and cytokine concentrations in PBMC-conditioned media after 24 h assessed via ELISA (**B**). CN = conditioned media from Control PBMCs without LPS stimulation; CS = conditioned media from Control PBMCs with LPS stimulation; KN = conditioned media from SCK PBMCs without LPS stimulation; KS = conditioned media from SCK PBMCs with LPS stimulation. For LPS, (+) indicates PBMCs stimulated with 10 ng/mL LPS during the 24 h culture period, while (−) indicates treatments without LPS. For SCK, (+) indicates PBMCs from SCK were used to make the conditioned media, while (−) indicates that PBMCs were derived from Control cows.

**Figure 4 animals-14-01995-f004:**
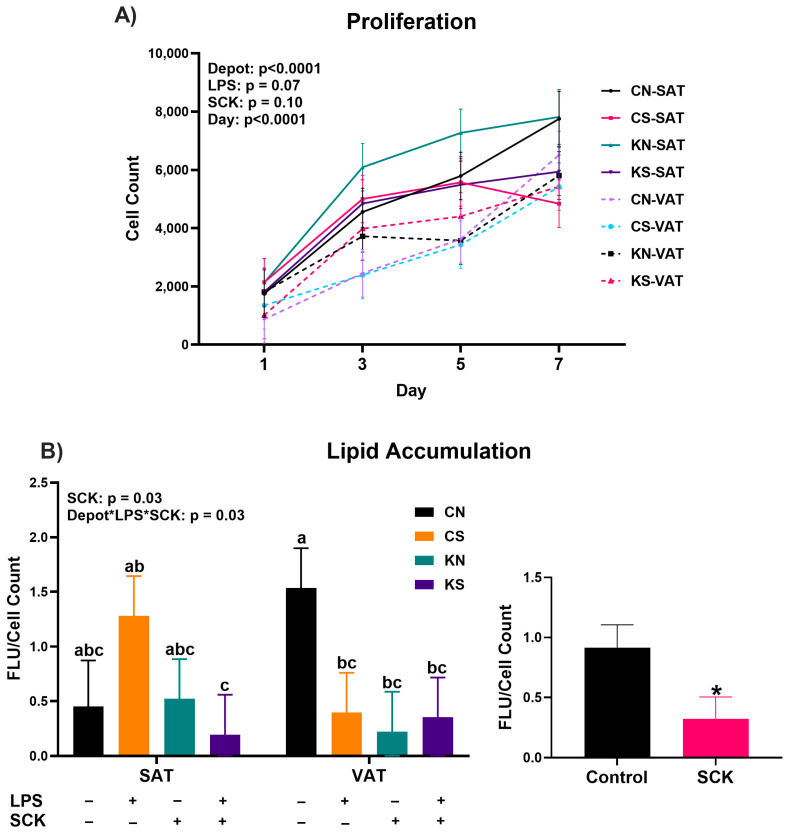
Preadipocyte proliferation (**A**) and mature adipocyte lipid accumulation (**B**) in cells treated with PBMC-conditioned media during the proliferation stage. Smaller graphs indicate the direction of the single effect (ex. Depot, LPS, SCK) or their interaction. SAT = subcutaneous adipose tissue preadipocytes; VAT = visceral adipose tissue preadipocytes; CN = conditioned media from Control PBMCs without LPS stimulation; CS = conditioned media from Control PBMCs with LPS stimulation; KN = conditioned media from SCK PBMCs without LPS stimulation; KS = conditioned media from SCK PBMCs with LPS stimulation. For LPS, (+) indicates PBMCs stimulated with 10 ng/mL LPS during the 24 h culture period, while (−) indicates treatments without LPS. For SCK, (+) indicates PBMCs from SCK were used to make the conditioned media, while (−) indicates that PBMCs were derived from Control cows. Differences between groups are denoted by letters (a,b,c) when interactions were identified as significant and by * for significant specific effects.

**Figure 5 animals-14-01995-f005:**
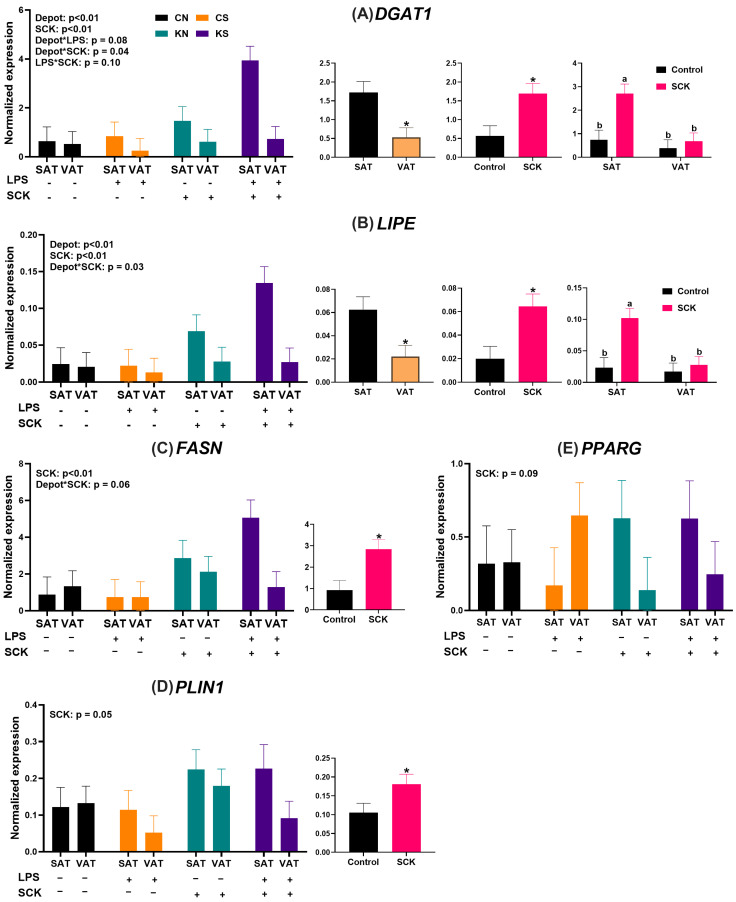
Expression of *DGAT1* (**A**), *LIPE* (**B**), *FASN* (**C**), *PLIN1* (**D**), and *PPARG* (**E**) in preadipocytes treated with PBMC-conditioned media for 48 h. Smaller graphs indicate the direction of the single effect (ex. Depot, LPS, SCK) or their interaction. SAT = subcutaneous adipose tissue preadipocytes; VAT = visceral adipose tissue preadipocytes. For LPS, (+) indicates PBMCs stimulated with 10 ng/mL LPS during the 24 h culture period, while (−) indicates treatments without LPS. For SCK, (+) indicates PBMCs from SCK were used to make the conditioned media, while (−) indicates that PBMCs were derived from Control cows. Differences between groups are denoted by letters (a,b) when interactions were identified as significant and by * for significant specific effects.

**Figure 6 animals-14-01995-f006:**
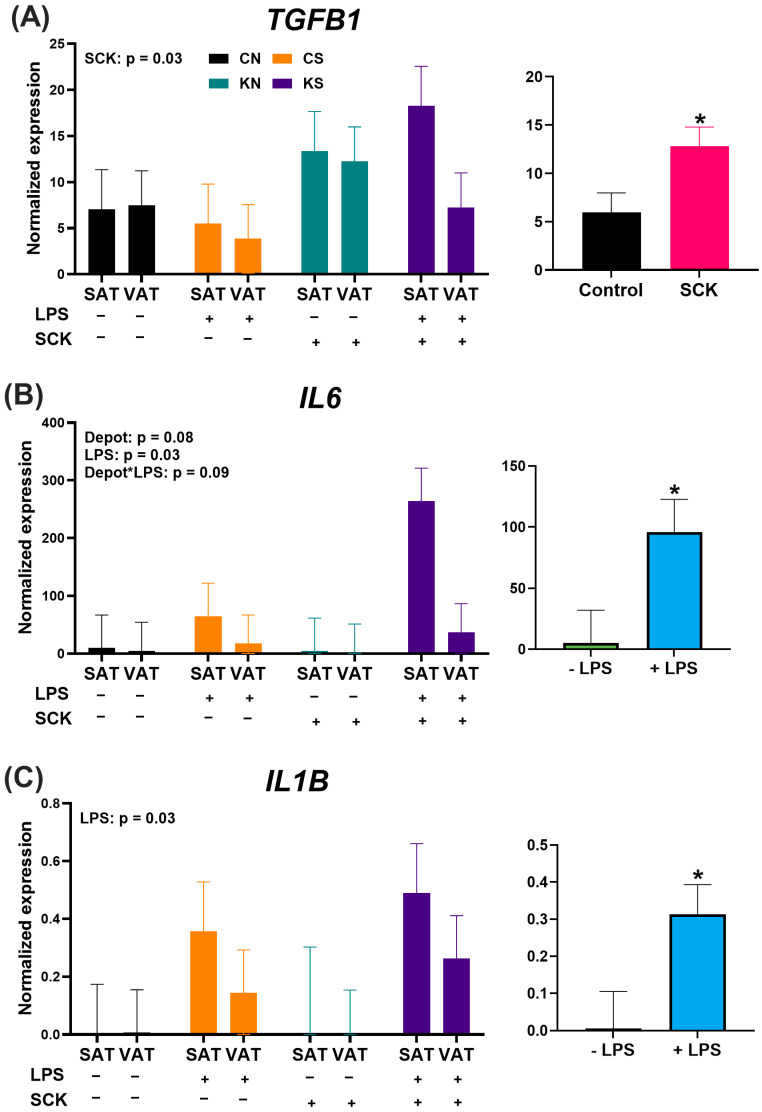
Expression of *TGFB1* (**A**), *IL6* (**B**), and *IL1B* (**C**) in preadipocytes treated with PBMC-conditioned media for 48 h. Smaller graphs indicate the direction of the single effect (ex. Depot, LPS, and SCK) or their interaction. SAT = subcutaneous adipose tissue preadipocytes; VAT = visceral adipose tissue preadipocytes. For LPS, (+) indicates PBMCs stimulated with 10 ng/mL LPS during the 24 h culture period, while (−) indicates treatments without LPS. For SCK, (+) indicates PBMCs from SCK were used to make the conditioned media, while (−) indicates that PBMCs were derived from Control cows. *: significant specific effects.

**Table 1 animals-14-01995-t001:** Comparison (mean ± SEM) of SCK and Control cows.

	SCK	Control	*p*-Value **
DIM	6.7 ± 3.8	8.1 ± 3.0	0.37
Lactations	3.2 ± 1.2	2.9 ± 1.4	0.62
BCS (1–5)	3.7 ± 0.4	3.8 ± 0.4	0.66
BHB (mM) *	1.5 ± 0.5	0.7 ± 0.1	<0.01

* Assessed using a Precision Xtra ketometer. ** Assessed using Student’s *t*-test in GraphPad Prism 10 (v.2.2).

## Data Availability

The original contributions presented in the study are included in the article/Appendix A, further inquiries can be directed to the corresponding author.

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
