# Peer review of "The Effect of Subclinical Ketosis on the Peripheral Blood Mononuclear Cell Inflammatory Response and Its Crosstalk with Depot-Specific Preadipocyte Function in Dairy Cows"

_animals, 2024, doi:10.3390/ani14131995_

Round 1

Reviewer 1 Report

Comments and Suggestions for Authors

The manuscript investigates the effects of Hyperketonemia on PBMC cells stimulated with LPS and the subsequent effects of this conditioned media on preadipocyte function in dairy cows. The topic is of strong interest to dairy Scientists involved in managing the transition period. Overall the manuscript is well written. I suggest revising the abstract and results section for better clarification

L16: Replace "metabolically stressed dairy cattle" with "hyperketonemic cows" or "cows with ketosis," since the authors do not evaluate other markers of metabolic stress.

L31: Correct "PMBC" to "PBMC" throughout the manuscript.

L33: include the BHB threshold in the abstract

L38: Clarify how the PBMCs were stimulated. Describe this before presenting the results to ensure the conditioned media description is clear.

L40-41: Clarify whether the higher expression of lipid metabolism-related genes was observed in preadipocytes from CON or SCK cows. Specify that preadipocytes were collected only from non-ketotic cows.  

L42: Indicate if these are differentiated adipocytes and specify the duration of differentiation. Clarify if these adipocytes came from CON or SCK cows and note any observed differences between the two groups. Also, specify whether PBMCs stimulated with LPS were collected from CON or SCK cows

Overall, the abstract will benefit of fine-tuning focusing on describing the stimulation of PBMCs

L54-56: Sordillo et al,. did not discuss any homeorhetic adaptations in that Lit Review. Consider a more appropriate reference. For example, Bauman and Currie 1980 were among the first authors to talk about homeorhetic in dairy cows

Materials and methods

L109: Specify the number of blocks created by DIM and parity. If cows were blocked by these categories, include this blocking effect in the statistical analysis.

Figure 1: I appreciate that the authors include this type of diagram for clarification.  Define abbreviations such as CUS, CS, SKUS, and SKS. Clarify if preadipocytes were from CON or SCK cows, preferably in the abstract and Figure 1. I was confused until I read the adipose tissue collection and I realized only preadipocytes from healthy cows were used. This distinction would be important before, for example in the abstract and figure 1.

L168-174: I suggest using a different nomenclature because it becomes confusing at some point to follow the graphs given the number of treatments and cell types. Especially because the letter “S” is present in many abbreviations. Consider using different nomenclature to avoid confusion due to the numerous treatments and cell types. For instance, use "Control Stimulated (CS)," "Ketosis Stimulated (KS)," "Control Non-Stimulated (CN)," and "Ketosis Non-Stimulated (KN)

L237: Clarify what the authors mean by "matched."

L272-282: Include whether any transformations were performed or if the data was normally distributed. Explain how the sample size was determined.

Figure 4 A is hard to follow. If possible, consider my recommendation about the abbreviations and reduce the size of the symbol to observe the continuous and dashed lines in the legend. If tendencies were declared < 0.1, clarify if LPS tended to decrease proliferation.

L329: The use of abbreviations is to reduce the complexity of a text. Reduce redundancy by using abbreviations consistently. For example, "VAT cells treated with CUS media had the highest lipid accumulation, and SAT cells treated with SKS media had the lowest (include CUS, SKS, CS, and SKUS in the graph 4 B to make it consistent with figure 3)

L332-334: Specify whether the higher lipid accumulation was observed in SAT or VAT. Also, refer to my previous comment about repeating the media explanation.

L352: This sentence needs further clarification. Clarify if "SCK conditioned media" refers to SKUS or SKS media and compare to which cells.

Figure 5 single effect is confusing because SCK and control in black and pink were used in Figure 2 to show the differences between cows with ketosis and healthy cows. Use different colors and abbreviations for the source of the media across the paper to avoid confusion. Consider calling this the "ketone effect" or another name to reduce confusion. Revise the results related to Figure 5 (Lines 349-369)

L384: I suggest a full revision of the manuscript focusing in improve the clarity of the description of the results. For example, sentence like this lacks clarity : “Expression of IL1B and IL6 was minimal in preadipocytes not treated with LPS (CUS and SKU) independent of ketosis status (Figure 6B,6C)”. adipocytes not treated with LPS sounds award. Preadipocytes treated with CUS and SKU media (however, these abbreviations are not in Figure 6B).

Discussion

L405-407: What is the rationale for stimulating PBMC with LPS in the present study? Did cows with subclinical ketosis present any marker of inflammation or endotoxemia? Is this cohort of SCK cows affected by inflammation or is this a finding specific to clinical ketosis?

Regarding my previous comment, authors should discuss and be aware that hyperketonemia is also a common finding in high-producing dairy cows and perhaps this is related to the lack of significant differences in PBMC. Check out this recent publication https://doi.org/10.1016/j.cvfa.2023.02.004

L409: Neutrophils are also WBCs.

L417: During in vitro studies…. reads better

L418: Discuss that low WBC, including neutrophils, could result from reduced proliferation or increased migration, as cows may have an active inflammatory status that is not detectable by physical examination. This limitation could be addressed if the authors had access to markers of inflammation in serum.

L447: Independent of which group? Is this ketotic status?

L451: Still struggling to understand if the conditioned media was SKUS or SKS. This demonstrates that the description of the results needs more work. Same in L454

Reviewer 2 Report

Comments and Suggestions for Authors

Dear authors,

thank you for submittin this interesting research paper paper into the Journal. we know that fresh cows are more sensitive for mastitis and other infections and this is another attempt to explain. The problem with smooth reading this manuscript is the use of soo many abbreviations and these must be explained in every table these are used. If not, you are losing a lot of potential interested readers. 

Considering table 1, it is clear that after calving you see in case of SCK in the 1st week an increase of NEFA en in the 2nd and 3rd an increase of BHB (see also Holzhauer and Valarcher, 2024). In Fig. 2 you do?? So why not here??

part 3.1. p-values with max 2 figures behind the comma, it is about 10 vs. 11 cows??

part 3.5 DGAT1?? etc. I have no idea :)

most of the discussion is OK, but with some abbreviations like in line 469, 499 etc. I am lost. 

in Fig 3 you should explain more the abbreviations, e.g. CXCL8, the meaning of?? I have no idea. and SKUS etc. ???   

Round 2

Reviewer 2 Report

Comments and Suggestions for Authors

a Reaction on your comment below:

part 3.1. p-values with max 2 figures behind the comma, it is about 10 vs. 11 cows??

Thank you for the comment. We are unsure as to how to interpret this revision. The p-values that are written out in the text are the exact p-values of the designated comparison and are meant to supplement the figures in which significance was only denoted with *. The sample size of each group has been added into the Figure 1 caption.

Reviewer

the p-values are not in line with the no. of animals used, so line 303 must be BHB (p< 0.01) .... NEFA; p=0.02) 

line 308: ... WBC (p=0.01;...) ,,, neutrophils (p=0.02; ...) 

Figure 2: D p=0.09

line 439: .... was not registered or noticed.  

Author Response

the p-values are not in line with the no. of animals used, so line 303 must be BHB (p< 0.01) .... NEFA; p=0.02) 

line 308: ... WBC (p=0.01;...) ,,, neutrophils (p=0.02; ...) 

Figure 2: D p=0.09

Thank you to the reviewer for the clarification. The p-values have been modified accordingly in the text as well as in Figures 2, 3, and 4. We greatly appreciate the reviewer for providing this feedback.

line 439: .... was not registered or noticed.  

We have replaced the text indicated on line 439 with what was suggested by the reviewer. Thank you for this valuable revision.